# MINI-BATCH SUBMODULAR MAXIMIZATION

## ABSTRACT

We present the first *mini-batch* algorithm for maximizing a non-negative monotone *decomposable* submodular function, $F = \sum_{i=1}^{N} f^i$, under a set of constraints. The expected number of oracle evaluations of our algorithm only depends on the size of the ground set. Previous results require a number of oracle evaluations that either depend on $N$ or have a worst-case *exponential* dependence on the size of the ground set.

## 1 INTRODUCTION

We consider the problem of maximizing a non-negative submodular function $F$. A set function $F : 2^E \to \mathbb{R}^+$ is submodular if for any subsets $S \subseteq T \subseteq E$ and $e \in E \setminus T$, it holds that

$$F(S + e) - F(S) \geq F(T + e) - F(T)$$

We focus on the case where $F$ is *decomposable*: $F = \sum_{i=1}^{N} f^i$, where each $f^i : 2^E \to \mathbb{R}^+$ is a non-negative submodular function on the ground set $E$ with $|E| = n$. Further, $\forall i, f^i$ is monotone ($\forall S \subseteq T \subseteq E, f^i(T) \geq f^i(S)$).

We assume that every $f^i$ is represented by an evaluation oracle that returns the value $f^i(S)$ for every $S \subseteq E$. Our goal is to maximize $F$ under some set of constraints while minimizing the number of oracle evaluations to $\{f^i\}$.

For $S, A \subseteq E$ we define $F_S(A) = F(S + A) - F(S)$. We slightly abuse notation and write $F_S(e), F(e)$ instead of $F_S(\{e\}), F(\{e\})$.

**Motivation and Background** For ease of presentation let us first focus on maximizing $F$ under a cardinality constraint $k$, i.e., $\max F(S), |S| \leq k$. The classical greedy algorithm (Nemhauser et al., 1978) (Algorithm 1) achieves an *optimal* $(1 - 1/e)$-approximation for this problem.

---

**Algorithm 1:** Greedy submodular maximization under a cardinality constraint

1  $S_1 \leftarrow \emptyset$
2  **for** $j = 1$ *to* $k$ **do**
3  $\quad\quad e' = \arg\max_{e \in E \setminus S_j} F_{S_j}(e)$
4  $\quad\quad S_{j+1} = S_j + e'$
5  **end**
6  return $S_{k+1}$

---

When $F$ is decomposable, and each evaluation of $f^i$ is counted as an oracle call, the above algorithm requires $O(Nnk)$ oracle calls. This can be prohibitively expensive if $N \gg n$. This raises the question: Can we eliminate the dependence on $N$?

Recently, Rafiey & Yoshida (2022) showed how to construct a *sparsifier* for $F$. That is, given a parameter $\epsilon > 0$ they show how to find a vector $w \in \mathbb{R}^N$ such that the number of non-zero elements in $w$ is small in expectation and the function $\hat{F} = \sum_{i=1}^{N} w_i f^i$ satisfies with high probability (w.h.p)[1]

---

[1]Probability at least $1 - 1/n^c$ for an arbitrary constant $c > 1$. The value of $c$ does affect the asymptotics of the results we state (including our own).

that

$$\forall S \subseteq E, (1 - \epsilon)F(S) \leq \hat{F}(S) \leq (1 + \epsilon)F(S)$$

Specifically, every $f^i$ is sampled with probability $\alpha_i$ proportional to $p_i = \max_{S \subseteq E, F(S) \neq 0} \frac{f^i(S)}{F(S)}$. If it is sampled, it is included in the sparsifier with weight $1/\alpha_i$, which implies that $\mathbb{E}[w_i] = 1$. While calculating the $p_i$'s exactly requires exponential time, Rafiey & Yoshida (2022) make do with an approximation, which can be calculated using interior point methods (Bai et al., 2016).

Specifically, Rafiey & Yoshida (2022) show that if all $f^i$'s are non-negative and monotone[2], the above sparsifier can be constructed by an algorithm that requires $poly(N)$ oracle evaluations and the sparsifier will have expected size $O(\epsilon^{-2}Bn^{2.5}\log n)$, where $B = \max_{i \in [N]} B_i$ and $B_i$ is the number of extreme points in the base polyhedron of $f^i$. They extend their results to matroid constraints of rank $r$ and show that a sparsifier with expected size $O(\epsilon^{-2}Brn^{1.5}\log n)$ can be constructed.

For the specific case of a cardinality constraint $k$, this implies a sparsifier of expected size $O(\epsilon^{-2}Bkn^{1.5}\log n)$ can be constructed using $poly(N)$ oracle evaluations. The sparsifier construction is treated as a *preprocessing step* (we elaborate on this in Section 1.1), and therefore the actual execution of Algorithm 1 on the sparsifier requires only $O(\epsilon^{-2}Bk^2n^{2.5}\log n)$ oracle evaluations to get a $(1 - 1/e - \epsilon)$ approximation. This is an improvement over Algorithm 1 when $N \gg n, B$.

Recently, Kudla & Zivný (2023) showed improved results for the case of *bounded curvature*. The *curvature* of a submodular function $F$ is defined as $c = 1 - \min_{S \subseteq E, e \in E \setminus S} \frac{F_S(e)}{F_\emptyset(e)}$. We say that $F$ has *bounded-curvature* if $c < 1$. Submodular functions with bounded curvature (Conforti & Cornuéjols, 1984) offer a balance between modularity and submodularity, capturing the essence of diminishing returns without being too extreme.

They show that when the curvature of all $f^i$'s and of $F$ is constant it is possible to reduce the preprocessing time to $O(Nn)$ oracle queries and to reduce the size of the sparsifier by a factor of $\sqrt{n}$. Furthermore, their results extend to the much more general case of *k-submodular functions*. While this significantly improves over the number of oracle calls compared to (Rafiey & Yoshida, 2022), the running time of the preprocessing step depends on $\log\left(\max_{i \in [N]} \frac{\max_{e \in E} f_\emptyset^i(e)}{\min_{e \in E, f_\emptyset^i(e) > 0} f_\emptyset^i(e)}\right)$.

There are two main issues with the above approach. The first is that, in general, constructing the sparsifier can be prohibitively slow. The second issue is the factor $B$ in the size of the sparsifier, which can be exponential in $n$. While Rafiey & Yoshida (2022) note that for some natural problems (e.g., facility location, maximum coverage), $B$ is small and the $p_i$'s can be computed efficiently, for general problems this can be a significant bottleneck.

## 1.1 OUR RESULTS

In this work, we focus on the *greedy algorithm* for constrained submodular maximization. We show that instead of sparsifying $F$, much better results can be achieved by using mini-batches during the execution of the greedy algorithm. That is, rather than sampling a large sparsifier $\hat{F}$ and performing the optimization process, we show that if we sample a much smaller sparsifier (a *mini-batch*), $\hat{F}^j$, for $j$-th step of the greedy algorithm, we can overcome both of the problems presented above. Specifically, our results are *independent* of $B$ and our preprocessing is extremely simple and only requires $O(nN)$ oracle evaluations.

While the mini-batch approach results in a significant improvement in performance, computing a sparsifier has the benefit of being independent of the algorithm. This means that we need to re-establish the approximation ratio of our mini-batch algorithm for different constraints. Although these proofs are often straightforward, compiling an exhaustive list of where the mini-batch method is applicable is both laborious and offers limited insights.

To illustrate the effectiveness of our method while maintaining readability, we focus on two widely researched constraints: the cardinality constraint and the $p$-system constraint (defined later in the section). The cardinality constraint was chosen for its simplicity and its prominence in research,

---

[2]Rafiey & Yoshida (2022) also present results for non-monotone functions, however, Kudla & Zivný (2023) point out an error in their calculation and note that the results only hold when all $f^i$'s are monotone.

while the $p$-system constraint was chosen for its broad applicability. We strongly believe that our approach could be applied beyond submodular functions (e.g., $k$-submodular functions, similar to Kudla & Zivný (2023)), achieve better approximation guarantees for specific constraints, and even applied beyond the greedy algorithm.

We compare our results with the results of Rafiey & Yoshida (2022); Kudla & Zivný (2023) and the naive algorithm (without sampling or sparsification) in Table 1[3]. While our results hold for the unbounded curvature case, we can get improved performance if the curvature is bounded. It's worth noting that while Kudla & Zivný (2023) assume every $f^i$ has bounded curvature, we only require $F$ to have bounded curvature.

| | Preprocessing | Cardinality constraint $(1 - 1/e - \epsilon)$-approx | $p$-system constraint $(\frac{1-\epsilon}{p+1})$-approx |
|---|---|---|---|
| Naive | None | $O(Nnk)$ | $O(Nnk)$ |
| Rafiey & Yoshida (unbounded curvature) | $poly(N)$ | $\widetilde{O}(B \cdot \frac{k^2 n^{2.5}}{\epsilon^2})$ | $\widetilde{O}(B \cdot \frac{k^2 n^{3.5}}{\epsilon^2})$ |
| **Our results** (unbounded curvature) | $O(Nn)$ | $\widetilde{O}(\frac{k^3 n^2}{\epsilon^2})$ | $\widetilde{O}(\frac{k^3 p^2 n^2}{\epsilon^2})$ |
| Kudla & Zivný (bounded curvature) | $O(Nn)$ | $\widetilde{O}(B \cdot \frac{k^2 n^2}{\epsilon^2})$ | $\widetilde{O}(B \cdot \frac{k^2 n^3}{\epsilon^2})$ |
| **Our results** (bounded curvature) | $O(Nn)$ | $\widetilde{O}(\frac{kn^2}{\epsilon^2})$ | $\widetilde{O}(\frac{kn^2}{\epsilon^2})$ |

Table 1: Comparison of the number of oracle queries during preprocessing and during execution. For ease of presentation and to allow comparison with Kudla & Zivný, we assume the curvature is constant. The preprocessing step of Rafiey & Yoshida uses interior point methods, therefore, it is significantly more costly than $O(Nn)$.

**Meta greedy algorithm**   Our starting point is the meta greedy algorithm (Algorithm 2). The algorithm executes for $k \leq n$ iterations where $k$ is some upper bound on the size of the solution. At every iteration, the set $A_j \subseteq E \setminus S_j$ represents some constraint that limits the choice of potential elements to extend $S_j$. The algorithm terminates either when the solution size reaches $k$ or when no further extensions to the current solution are possible (i.e., $A_j = \emptyset$). Furthermore, the algorithm does not have access to the exact *incremental oracle*, $F_{S_j}$, at every iteration, but only to some approximation.

---

**Algorithm 2:** Meta greedy algorithm with an approximate oracle

1  $S_1 \leftarrow \emptyset$
2  Let $k$ be an upper bound on the size of the solution
3  **for** $j = 1$ *to* $k$ **do**
4  $\quad$ Let $A_j \subseteq E \setminus S_j$ $\qquad\qquad\qquad\qquad$ ▷ Problem specific constraint
5  $\quad$ **if** $A_j = \emptyset$ **then** return $S_j$
6  $\quad$ Let $\hat{F}^j_{S_j}$ be an approximation for $F_{S_j}$ $\qquad$ ▷ Problem specific approximation
7  $\quad$ $e_j = \arg\max_{e \in A_j} \hat{F}^j_{S_j}(e)$
8  $\quad$ $S_{j+1} = S_j + e_j$
9  **end**
10 return $S_{k+1}$

---

Before we formally define "approximation" in the above, let us note that when we have access to exact values of $F_{S_j}$, Algorithm 2 captures many variants of the greedy submodular maximization algorithm. For example, setting $A_j = E \setminus S_j$ we get the algorithm of Nemhauser et al. (1978) for maximizing a non-negative submodular function under a cardinality constraint. This meta-algorithm also captures the case of maximization under a *p-system constraint*.

---

[3]Where $\widetilde{O}$ hides $\log n$ factors.

$p$-**systems** The concept of $p$-systems offers a generalized framework for understanding independence families, parameterized by an integer $p$. We can define a $p$-system in the context of an independence family $\mathcal{I} \subseteq 2^E$ and $E' \subseteq E$. Let $\mathcal{B}(E')$ be the maximal independent sets within $\mathcal{I}$ that are also subsets of $E'$. Formally,

$$\mathcal{B}(E') = \{A \in \mathcal{I} | A \subseteq E' \text{ and no } A' \in \mathcal{I} \text{ exists such that } A \subset A' \subseteq E'\}.$$

A distinguishing characteristic of a $p$-system is that for every $E' \subseteq E$, the ratio of the sizes of the largest to the smallest sets in $\mathcal{B}(E')$ does not exceed $p$:

$$\frac{\max_{A \in \mathcal{B}(E')} |A|}{\min_{A \in \mathcal{B}(E')} |A|} \leq p.$$

The significance of $p$-systems lies in their ability to encapsulate a variety of combinatorial structures. For instance, when we consider the intersection of $p$ matroids, they can be aptly described using $p$-systems. To provide more tangible examples, in graph theory, the collection of matchings in a standard graph can be viewed as a 2-system. Extending this to hypergraphs, where edges might have cardinalities up to $p$, the set of matchings therein can be conceptualized as a $p$-system.

**The greedy algorithm for $p$-systems** Formally, the optimization problem can be expressed as: $\max_{S \in \mathcal{I}} F(S)$ where the pair $(E, \mathcal{I})$ characterizes a $p$-system and $F : 2^E \to \mathbb{R}^+$ denotes a non-negative monotone submodular set function. It was shown by Nemhauser et al. (1978) that the natural greedy approach achieves an optimal approximation ratio of $\frac{1}{p+1}$. Setting $A_j = \{e \mid S_j + e \in \mathcal{I}\}$ (i.e., $S_j$ remains an independent set after adding $e$) in Algorithm 2 we get the greedy algorithm of Nemhauser et al. (1978). Note that for general $p$-systems it might be that $k = n$, however, there are very natural problems where $k \ll n$. For example, for maximum matching $E$ corresponds to all edges in the graph, which can be quadratic in the number of nodes, while the solution is at most linear in the number of nodes.

**Approximate oracles** In many scenarios we do not have access to *exact* values of $F_{S_j}$, and instead we must make do with an approximation. We start with the notion of an *approximate incremental oracle* introduced in (Goundan & Schulz, 2007). We say that $\hat{F}_{S_j}^j$ is an $(1 - \epsilon)$-approximate incremental oracle if

$$\forall e \in A_j, (1 - \epsilon)F_{S_j}(e) \leq \hat{F}_{S_j}^j(e) \leq (1 + \epsilon)F_{S_j}(e)$$

It was shown in (Goundan & Schulz, 2007; Călinescu et al., 2011)[4] that given a $(1 - \epsilon)$-approximate incremental oracle, the greedy algorithm under both a cardinality constraint and a $p$-system constraint achieves almost the same (optimal) approximation ratio as the non-approximate case.

**Theorem 1.** *Algorithm 2 with an $(1 - \epsilon)$-approximate incremental oracle has the following guarantees w.h.p.*

- *It achieves a $(1 - 1/e - \epsilon)$-approximation under a cardinality constraint $k$ (Goundan & Schulz, 2007).*

- *It achieves a $\left(\frac{1-\epsilon}{1+p}\right)$-approximation under a $p$-system constraint (Călinescu et al., 2011).*

We introduce a weaker type of approximate incremental oracle, which we call an *additive* approximate incremental oracle. We extend the results of Theorem 1 for this case. Let $S^*$ be some optimal solution for $F$ (under the relevant set of constraints). We say that $\hat{F}_{S_j}^j$ is an *additive* $\epsilon'$-approximate incremental oracle if

$$\forall e \in A_j, F_{S_j}(e) - \epsilon'F(S^*) \leq \hat{F}_{S_j}^j \leq F_{S_j}(e) + \epsilon'F(S^*)$$

This might seem problematic at first glance, as it might be the case that $F(S^*) \gg F_{S_j}(e)$. Luckily, the proofs guaranteeing the approximation ratio are *linear* in nature. Therefore, by the end of the proof we end up with an expression of the form:

$$F(S_{k+1}) \geq F(S^*)\beta + \gamma\epsilon'F(S^*)$$

---

[4]Strictly speaking, both Goundan & Schulz (2007) and Călinescu et al. (2011) define the approximate incremental oracle to be a function that returns $e_j$ at iteration $j$ of the greedy algorithm such that $\forall e \in A_j, F_{S_j}(e_j) \geq (1 - \epsilon)F_{S_j}(e)$. Our definition guarantees this property while allowing easy analysis of the mini-batch algorithm.

Where $\beta$ is the desired approximation ratio and $\gamma$ depends on the parameters of the problem (e.g., $\beta = (1 - 1/e), \gamma = 2k$ for a cardinality constraint). We can achieve the desired result by setting $\epsilon' = \epsilon/\gamma$. We state the following theorem (the proofs are very similar to those of Goundan & Schulz (2007); Călinescu et al. (2011), and we defer them to the Appendix).

**Theorem 2.** *Algorithm 2 with an additive $\epsilon'$-approximate incremental oracle has the following guarantees w.h.p.*

- *If $\epsilon' < \epsilon/2k$, it achieves a $(1 - 1/e - \epsilon)$-approximation under a cardinality constraint $k$.*

- *If $\epsilon' < \epsilon/2kp$, it achieves a $(\frac{1-\epsilon}{1+p})$-approximation under a $p$-system constraint.*

**Mini-batch sampling** Our main result shows that when $\hat{F}_{S_j}^j$ is sampled using mini-batch sampling we indeed get, w.h.p, an (additive) approximate incremental oracle for every step of the algorithm. We present our sampling procedure in Algorithm 3. It takes in a batch size parameter $\alpha$ and samples every $f^i$ with probability proportional to $\alpha p_i$. The main benefit in our approach is that its is sufficient to set $p_i = \max_{e \in E, F_\emptyset(e) \neq 0} \frac{f_\emptyset^i(e)}{F_\emptyset(e)}$ compared to $\max_{S \subseteq E, F(S) \neq 0} \frac{f^i(S)}{F(S)}$ in (Rafiey & Yoshida, 2022). This only requires $O(Nn)$ oracle evaluations.

Similar to Rafiey & Yoshida (2022) we treat the computation of the $p_i$'s as a preprocessing step. The justification for this, is that the $p_i$'s do not depend on the *constraints* of the problem. Therefore, computing the $p_i$'s a single time, we can execute our algorithm on various constraints (e.g., different $p$-systems).

---

**Algorithm 3:** Sample($\alpha$)

1   $\forall i \in [N], p_i \leftarrow \max_{e \in E, F_\emptyset(e) \neq 0} \frac{f_\emptyset^i(e)}{F_\emptyset(e)}$        ▷ Computed once, during preprocessing
2   $w \leftarrow 0$
3   **for** $i = 1$ *to* $N$ **do**
4      $\alpha_i \leftarrow \min\{1, \alpha p_i\}$
5      $w_i \leftarrow 1/\alpha_i$ with probability $\alpha_i$        ▷ Do nothing with probability $1 - \alpha_i$
6   **end**
7   return $\hat{F} = \sum_{i=1}^N w_i f^i$

---

Plugging Algorithm 3 into Line 6 of Algorithm 2 we get our *mini-batch greedy algorithm*. That is, in the $j$-th iteration, we call Algorithm 3, get back $\hat{F}$ and set $\hat{F}_{S_j}^j(e) = \hat{F}_{S_j}(e)$.

In Section 2 we analyze the relation between the batch parameter, $\alpha$, and the the type of approximate incremental oracles guaranteed by our sampling procedure. We state the main theorem for the section below.

**Theorem 3.** *The mini-batch greedy algorithm maximizing a non-negative monotone submodular function has the following guarantees:*

- *If $F$ has curvature bounded by $c$, and $\alpha = \Theta(\frac{\log n}{\epsilon^2(1-c)})$ it holds w.h.p that $\forall j \in [k]$ that $\hat{F}_{S_j}^j$ is an $(1 - \epsilon)$-approximate incremental oracle.*

- *If $\alpha = \Theta(\epsilon^{-2} \log n)$ it holds w.h.p that $\forall j \in [k]$ that $\hat{F}_{S_j}^j$ is an additive $\epsilon$-approximate incremental oracle.*

*Furthermore, the number of oracle evaluations during preprocessing is $O(nN)$ and an expected $\alpha(\sum_{i=1}^N p_i)(\sum_{j=1}^k |A_j|) = O(\alpha k n^2)$ during execution.*

Combining Theorem 3 with Theorem 1 and Theorem 2 we state our main result.

**Theorem 4.** *The mini-batch greedy algorithm maximizing a non-negative monotone submodular function requires $O(nN)$ oracle calls during preprocessing and has the following guarantees:*

- *If $F$ has curvature bounded by $c$, it achieves w.h.p a $(1 - 1/e - \epsilon)$-approximation under a cardinality constraint and $(\frac{1-\epsilon}{1+p})$-approximation under a $p$-system constraint with an expected $O(\frac{kn^2 \log n}{\epsilon^2(1-c)})$ oracle evaluations for both cases.*

- *It achieves w.h.p a $(1 - 1/e - \epsilon)$-approximation under a cardinality constraint and $(\frac{1-\epsilon}{1+p})$-approximation under a $p$-system constraint with an expected $O(k^3(n/\epsilon)^2 \log n)$ and $O(k^3(pn/\epsilon)^2 \log n)$ oracle evaluations respectively.*

## 1.2 RELATED WORK

**Approximate oracles** Apart from the results of (Goundan & Schulz, 2007; Călinescu et al., 2011) there are works that use different notions of an approximate oracle. Several works consider an approximate oracle $\hat{F}$, such that $\forall S \subseteq E, \left|\hat{F}(S) - F(S)\right| < \epsilon F(S)$ (Crawford et al., 2019; Horel & Singer, 2016; Qian et al., 2017). The main difference of these models to our work is the fact that they do not assume the surrogate function, $\hat{F}$, to be submodular. This adds a significant complication to the analysis and degrades the performance guarantees.

**Mini-batch methods** The closest results resembling mini-batch methods for submodular functions are due to Buchbinder et al. (2015); Mirzasoleiman et al. (2015). They improve the expected query complexity of the greedy algorithm under a cardinality constraint by only considering a small random subset of $E \setminus S_j$ at the $j$-th iteration. We note that their approach can be combined into our mini-batch algorithm, reducing our query complexity by a $\tilde{\Theta}(k)$ factor, resulting in an approximation guarantee in expectation instead of w.h.p.

**Decomposable submodular functions** An excellent survey of the importance of decomposable functions is given in Rafiey & Yoshida (2022), which we summarize below. Decomposable submodular functions are prevalent in both machine learning and economic studies. In economics, they play a pivotal role in welfare optimization during combinatorial auctions (Dobzinski & Schapira, 2006; Feige, 2009; Feige & Vondrák, 2006; Papadimitriou et al., 2008; Vondrák, 2008). In machine learning, these functions are instrumental in tasks like data summarization, aiming to select a concise yet representative subset of elements. Their utility spans various domains, from exemplar-based clustering by (Dueck & Frey, 2007) to image summarization (Tschiatschek et al., 2014), recommender systems (Parambath et al., 2016) and document summarization (Lin & Bilmes, 2011). The optimization of these functions, especially under specific constraints (e.g., cardinality, matroid) has been studied in various data summarization settings (Mirzasoleiman et al., 2016a;b;c) and differential privacy (Chaturvedi et al., 2021; Mitrovic et al., 2017; Rafiey & Yoshida, 2020).

## 2 ANALYSIS OF THE MINI-BATCH GREEDY ALGORITHM

Let us start by bounding the expected size of $\hat{F}$ in Algorithm 3. We start with the following useful lemma.

**Lemma 5.** *It holds that $\sum_{i=1}^{N} p_i \leq n$.*

*Proof.* Let us divide the range $[N]$ into $A_e = \left\{i \in N \mid e = \arg\max_{e' \in E} \frac{f_\emptyset^i(e')}{F_\emptyset(e')}\right\}$. If 2 elements in $E$ achieve the maximum value for some $i$, we assign it to a single $A_e$ arbitrarily.

$$\sum_{i=1}^{N} p_i = \sum_{i=1}^{N} \max_{e \in E} \frac{f_\emptyset^i(e)}{F_\emptyset(e)} = \sum_{e \in E} \sum_{i \in A_e} \frac{f_\emptyset^i(e)}{F_\emptyset(e)} = \sum_{e \in E} \frac{\sum_{i \in A_e} f_\emptyset^i(e)}{F_\emptyset(e)} \leq \sum_{e \in E} 1 \leq n$$

$\square$

Using the above we state the following lemma:

**Lemma 6.** *The expected size of $\hat{F}$ is $\alpha \sum_{i=1}^{N} p_i \leq \alpha n$.*

*Proof.* Let $X_i$ be an indicator variable for the event $w_i > 0$. We are interested in $\sum_{i=1}^{N} \mathbb{E}[X_i]$. It holds that:

$$\sum_{i=1}^{N} \mathbb{E}[X_i] = \sum_{i=1}^{N} \alpha_i \le \sum_{i=1}^{N} \alpha p_i = \alpha \sum_{i=1}^{N} p_i \le \alpha n$$

$\square$

Next, let us show that $\hat{F}$ returned by Algorithm 3 is indeed an (additive) approximate incremental oracle w.h.p. We make use of the following Hoeffding bound.

**Theorem 7** (Hoeffding bound). *Let $X_1, ..., X_N$ be independent random variables in the range $[0, a]$. Let $X = \sum_{i=1}^{N} X_i$. Then for any $\epsilon \in [0, 1]$ and $\mu \ge \mathbb{E}[T]$,*

$$\mathbb{P}(|X - \mathbb{E}[X]| \ge \epsilon \mu) \le 2 \exp\left(-\frac{\epsilon^2 \mu}{3a}\right)$$

The following lemma provides concentration guarantees for $\hat{F}$ in Algorithm 3.

**Lemma 8.** *For every $S \subseteq E$ ($\hat{F}$ sampled after $S$ is fixed) and for every $e \in E$ and $\mu \ge F_S(e)$, it holds that*

$$\mathbb{P}\left[|\hat{F}_S(e) - F_S(e)| \ge \epsilon \mu\right] \le 2 \exp\left(-\frac{\epsilon^2 \mu}{3 F_\emptyset(e)/\alpha}\right)$$

*Proof.* Fix some $e \in E$. Let $G = \sum_{i \in I} f^i$, where $I = \{i \in [N] \mid \alpha_i = 1\}$. Let $F'_S(e) = F_S(e) - G_S(e)$ and $\hat{F}'_S(e) = \hat{F}_S(e) - G_S(e)$. Let $J = [N] \setminus I$. It holds that:

$$\mathbb{P}\left[|\hat{F}_S(e) - F_S(e)| \ge \epsilon \mu\right] = \mathbb{P}\left[|\hat{F}'_S(e) + G_S(e) - F'_S(e) - G_S(e)| \ge \epsilon \mu\right] = \mathbb{P}\left[|\hat{F}'_S(e) - F'_S(e)| \ge \epsilon \mu\right]$$

Due to the fact that $\mathbb{E}[w_i] = 1$ we have $\mathbb{E}[\hat{F}'_S(e)] = \mathbb{E}[\sum_{i \in J} w_i f^i_S(e)] = F'_S(e)$. As $f^i$'s are monotone, it holds that $\mu \ge F_S(e) \ge F'_S(e)$. Applying a Hoeffding bound (Theorem 7) we have

$$\mathbb{P}\left[|\hat{F}'_S(e) - F'_S(e)| \ge \epsilon \mu\right] \le 2 \exp\left(-\epsilon^2 \mu / 3a\right)$$

where $a = \max\{w_i f^i_S(e)\}_{i \in J}$. Recall that $w_i = 1/\alpha_i$ where $\alpha_i = \min\{1, \alpha p_i\}$ and $\alpha_i < 1$ for all $i \in J$. Let us upper bound $a$.

$$a = \max_{i \in J} w_i f^i_S(e) = \max_{i \in J} \frac{f^i_S(e)}{\alpha p_i} = \max_{i \in J} \frac{f^i_S(e)}{\alpha \cdot \max_{e' \in E} \frac{f^i_S(e')}{F_\emptyset(e')}} \le \max_{i \in J} \frac{f^i_\emptyset(e)}{\alpha \cdot \frac{f^i_\emptyset(e)}{F_\emptyset(e)}} = \frac{F_\emptyset(e)}{\alpha}$$

Where the inequality is due to submodularity and non-negativity in the nominator and maximality in the denominator. Given the above upper bound for $a$ we get:

$$\mathbb{P}\left[|\hat{F}_S(e) - F_S(e)| \ge \epsilon \mu\right] \le 2 \exp\left(-\epsilon^2 \mu / 3a\right) \le 2 \exp\left(-\frac{\epsilon^2 \alpha \mu}{3 F_\emptyset(e)}\right)$$

$\square$

Using the above we state the main theorem for this section.

**Theorem 3.** *The mini-batch greedy algorithm maximizing a non-negative monotone submodular function has the following guarantees:*

- *If $F$ has curvature bounded by $c$, and $\alpha = \Theta(\frac{\log n}{\epsilon^2(1-c)})$ it holds w.h.p that $\forall j \in [k]$ that $\hat{F}^j_{S_j}$ is an $(1 - \epsilon)$-approximate incremental oracle.*

- *If $\alpha = \Theta(\epsilon^{-2} \log n)$ it holds w.h.p that $\forall j \in [k]$ that $\hat{F}^j_{S_j}$ is an additive $\epsilon$-approximate incremental oracle.*

*Furthermore, the number of oracle evaluations during preprocessing is $O(nN)$ and an expected $\alpha(\sum_{i=1}^N p_i)(\sum_{j=1}^k |A_j|) = O(\alpha k n^2)$ during execution.*

*Proof.* The number of oracle evaluations is due to Lemma 5 and the fact that the algorithm executes for $k$ iteration and must evaluate $|A_j| \leq n$ elements per iteration.

Let us prove the approximation guarantees. Let us start with the bounded curvature case. Fix some $S_j$. As $\hat{F}^j$ is sampled after $S_j$ is fixed, we can fix some $e \in E$ and apply Lemma 8 with $\mu = F_{S_j}(e)$. We get that:

$$\mathbb{P}\left[|\hat{F}^j_{S_j}(e) - F_{S_j}(e)| \geq \epsilon F_{S_j}(e)\right] \leq 2 \exp\left(-\frac{\epsilon^2 \alpha F_{S_j}(e)}{3 F_\emptyset(e)}\right) \leq 2 \exp\left(-\frac{\epsilon^2 \alpha (1-c)}{3}\right) \leq 1/n^3$$

Where the second inequality is due to the fact that $F_{S_j}(e)/F_\emptyset(e) \geq \min_{S \subseteq E, e' \in E \setminus S} F_S(e')/F_\emptyset(e') = 1 - c$, and the last transition is by setting an appropriate constant in $\alpha = \Theta(\frac{\log(n)}{\epsilon^2(1-c)})$.

When the curvature is not bounded, we set $\mu = F_\emptyset(e) \geq F_{S_j}(e)$ and get:

$$\mathbb{P}\left[|\hat{F}^j_{S_j}(e) - F_{S_j}(e)| \geq \epsilon F_\emptyset(e)\right] \leq 2 \exp\left(-\frac{\epsilon^2 \alpha F_\emptyset(e)}{3 F_\emptyset(e)}\right) \leq 2 \exp\left(-\frac{\epsilon^2 \alpha}{3}\right) \leq 1/n^3$$

Where again the last inequality is by setting an appropriate constant in $\alpha = \Theta(\epsilon^{-2} \log n)$.

For both cases, we take a union bound over all $e \in E$ and $j \in [k]$ (at most $n^2$ values), which concludes the proof. $\qquad \square$

Note that in the above we use the fact that $F_\emptyset(e) \leq F(S^*)$ to get the second result. This is sufficient for our proofs to go through, however, the theorem has a much stronger guarantee which might be useful in other contexts.

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

## A    PROOF OF THEOREM 2

Theorem 2 directly follows from the two lemmas below.

**Lemma 9.** *Let $\epsilon' \leq \epsilon/2k$. Algorithm 2 with an additive $\epsilon'$-approximate incremental oracle achieves a $(1 - 1/e - \epsilon)$-approximation under a cardinality constraint $k$.*

*Proof.* Let $S^*$ be some optimal solution for $F$. We start by proving that the following holds for every $j \in [k]$:

$$F(S_{j+1}) - F(S_j) \geq \frac{1}{k}((1 - \epsilon)F(S^*) - F(S_j))$$

Fix some $j \in [k]$ and let $S^* \setminus S_j = \{e_1^*, \ldots, e_\ell^*\}$ where $\ell \leq k$. Let $S_t^* = \{e_1^*, \ldots, e_t^*\}$, and $S_0^* = \emptyset$. Let us first use submodularity and monotonicity to upper bound $F(S^*)$.

$$F(S^*) \leq F(S^* + S_j) = F(S_j) + \sum_{t=1}^{\ell}[F(S_j + S_t^*) - F(S_j + S_{t-1}^*)]$$

$$\leq F(S_j) + \sum_{t=1}^{\ell} F_{S_j}(e_t^*) \leq F(S_j) + \sum_{t=1}^{\ell} \max_{e \in E \setminus S_j} F_{S_j}(e)$$

$$\leq F(S_j) + k \max_{e \in E \setminus S_j} F_{S_j}(e) \leq F(S_j) + k(\max_{e \in E \setminus S_j} \hat{F}_{S_j}^j(e) + \epsilon' F(S^*))$$

Where the last inequality is due to the fact that $\hat{F}_{S_j}^j$ is an additive $\epsilon'$-approximate incremental oracle.

Noting that $e_j = \arg\max_{e \in E \setminus S_j} \hat{F}_{S_j}^j(e)$ we get that:

$$F(S^*) \leq F(S_j) + k(\hat{F}_{S_j}^j(e_j) + \epsilon' F(S^*))$$

$$\implies \hat{F}_{S_j}^j(e_j) \geq \frac{1}{k}((1 - \epsilon'k)F(S^*) - F(S_j))$$

The above lower bounds the progress on the $j$-th mini-batch. Now, let us bound the progress on $F$. Again, we use the fact that $\hat{F}_{S_j}^j$ is an additive $\epsilon'$-approximate incremental oracle.

$$F(S_{j+1}) - F(S_j) \geq \hat{F}_{S_j}^j(e_j) - \epsilon' F(S^*)$$

$$\geq \frac{1}{k}((1 - \epsilon'k)F(S^*) - F(S_j)) - \epsilon' F(S^*) \geq \frac{1}{k}((1 - 2\epsilon'k)F(S^*) - F(S_j))$$

Finally, using the fact that $\epsilon' \leq \epsilon/2k$ we get:

$$F(S_{j+1}) - F(S_j) \geq \frac{1}{k}((1 - \epsilon)F(S^*) - F(S_j))$$

Rearranging, the result directly follows using standard arguments.

$$F(S_{k+1}) > \frac{(1-\epsilon)}{k}F(S^*) + (1 - \frac{1}{k})F(S_k) \geq \frac{(1-\epsilon)}{k}F(S^*)(\sum_{i=0}^{k}(1 - \frac{1}{k})^i) + F(\emptyset)$$

$$\geq F(S^*)\frac{(1-\epsilon)(1 - \frac{1}{k})^k}{k(1 - (1 - \frac{1}{k}))} = (1 - \epsilon)(1 - \frac{1}{k})^k F(S^*) \geq (1 - \epsilon)(1 - 1/e)F(S^*) \geq (1 - 1/e - \epsilon)F(S^*)$$

$$\square$$

**Lemma 10.** *Let $\epsilon' \leq \epsilon/2kp$. Algorithm 2 with an additive $\epsilon'$-approximate incremental oracle achieves a $(\frac{1-\epsilon}{1+p})$-approximation under a $p$-system constraint.*

*Proof.* Let $S^*$ be some optimal solution for $F$. Assume without loss of generality that the solution returned by the algorithm consists of $k$ elements $S_{k+1} = \{e_1, \ldots, e_k\}$.

We show the existence of a partition $S_1^*, S_2^*, \ldots, S_k^*$ of $S^*$ such that $F_{S_j}(e_j) \geq \frac{1}{p} F_{S_{k+1}}(S_j^*) - 2\epsilon' F(S^*)$. Note, we allow some of the sets in the partition to be empty.

Define $T_k = S^*$. For $j = k, k-1, ..., 2$ execute: Let $B_j = \{e \in T_j \mid S_j + e \in \mathcal{I}\}$. If $|B_j| \leq p$ set $S_j^* = B_j$; else pick an arbitrary $S_j^* \subset B_j$ with $|S_j^*| = p$. Then set $T_{j-1} = T_j \setminus S_j^*$ before decreasing $j$. After the loop set $S_1^* = T_1$. It is clear that for $j = 2, ..., k$, $|S_j^*| \leq p$.

We prove by induction over $j = 0, 1, ..., k-1$ that $|T_{k-j}| \leq (k-j)p$. For $j = 0$, when the greedy algorithm stops, $S_{k+1}$ is a maximal independent set contained in $E$, therefore any independent set (including $T_k = S^*$) satisfies $|T_k| \leq p|S_{k+1}| = pk$. We proceed to the inductive step for $j > 0$. There are two cases: (1) $|B_{k-j+1}| > p$, which implies that $\left|S_{k-j+1}^*\right| = p$ and using the induction hypothesis we get that $|T_{k-j}| = |T_{k-j+1}| - \left|S_{k-j+1}^*\right| \leq (k-j+1)p - p = (k-j)p$. (2) $|B_{k-j+1}| \leq p$, it holds that $T_{k-j} = T_{k-j+1} \setminus B_{k-j+1}$. Let $Y = S_{k-j+1} + T_{k-j}$. Due to the definition of $B_{k-j+1}$ it holds that $S_{k-j+1}$ is a maximal independent set in $Y$. It holds that $T_{k-j}$ is independent and contained in $Y$, therefore $|T_{k-j}| \leq p|S_{k-j+1}| = p(k-j)$.

Finally, we get that $|T_1| = |S_1^*| \leq p$. By construction it holds that $\forall j \in [k], \forall e \in S_j^*, S_j + e$ is independent. From the choice made by the greedy algorithm and the fact that $\hat{F}_{S_j}^j$ is an additive $\epsilon'$-approximate incremental oracle it follows that for each $e \in S_j^*$:

$$F_{S_j}(e_j) \geq \hat{F}_{S_j}^j(e_j) - \epsilon' F(S^*) \geq \hat{F}_{S_j}^j(e) - \epsilon' F(S^*) \geq F_{S_j}(e) - 2\epsilon' F(S^*)$$

Hence,

$$\left|S_j^*\right| F_{S_j}(e_j) \geq \sum_{e \in S_j^*} (F_{S_j}(e) - 2\epsilon' F(S^*)) \geq F_{S_j}(S_j^*) - 2\epsilon' \left|S_j^*\right| F(S^*) \geq F_{S_{k+1}}(S_j^*) - 2\epsilon' \left|S_j^*\right| F(S^*)$$

Using submodularity in the last two inequalities.

For all $j \in \{1, 2, ..., k\}$ it holds that $\left|S_j^*\right| \leq p$, and thus $F_{S_j}(e_j) \geq \frac{1}{p} F_{S_{k+1}}(S_j^*) - 2\epsilon' F(S^*)$. Using the partition we get that:

$$F(S_{k+1}) \geq \sum_{j=1}^{k} F_{S_j}(e_j) \geq \sum_{j=1}^{k} \left(\frac{1}{p} F_{S_{k+1}}(S_j^*) - 2\epsilon' F(S^*)\right)$$

$$\geq \frac{1}{p} F_{S_{k+1}}(S^*) - 2\epsilon' k F(S^*) \geq \frac{1}{p}(F(S^*) - F(S_{k+1})) - 2\epsilon' k F(S^*)$$

Where the second to last inequality is due to submodularity and the last is due to monotonicity. Rearranging we get that:

$$F(S_{k+1}) \geq \frac{(1 - 2p\epsilon' k)}{p+1} F(S^*)$$

As $\epsilon' < \frac{\epsilon}{2pk}$ we get the desired result. □

