# OpenReview forum: "Mini-batch Submodular Maximization"
_ICLR.cc/2024/Conference — ICLR 2024 Conference Withdrawn Submission_

### Official Review · Reviewer_qFjC · 2023-10-25

**Soundness:** 2 fair
**Presentation:** 1 poor
**Contribution:** 2 fair
**Rating:** 5
**Confidence:** 4

**Summary:**

The paper studies maximizing a submodular function F that is decomposable into N submodular functions. It provides a method to sample a subset of these N functions and form a weighted sum to approximate F. They show that with enough samples, they can get the optimal guarantees of the Greedy algorithm for cardinality constraint and also p-systems. Their algorithm becomes better than the naive approach if N is larger than n times k where n is the number of items in the ground set and k is the cardinality constraint.

**Strengths:**

They provide a sampling method to approximate decomposable function F with a weighted sum of its member functions such that the number of terms is independent of N. Here N is the number of submodular functions contributing to F. The number of terms depend polynomially on the size of the ground set.

**Weaknesses:**

Their algorithm becomes better than the naive approach if N is larger than n times k where n is the number of items in the ground set and k is the cardinality constraint.

Question to authors: Could you provide a practical example that N is substantially larger than n times k to motivate your algorithms?

Approximation of function F:
In line 6 of algorithm 2, you look for an approximate version of the function F_{S_j}. The description of how you use 3 to achieve that is not completely clear. But the likeliest interpretation that comes to mind is the following. You call algorithm 3 with function F initialized as F_{S_j} in the j’th iteration of the Greedy algorithm. This requires O(Nn) oracle calls because of line 1 of Alg 3 every time you call Alg 3. Since you need k approximate functions for the k iterations of the greedy algorithm, overall you have O(Nnk) oracle calls which makes the whole motivation point of your paper irrelevant.

The statement of Lemma 8 suggests that you had the above interpretation in mind (\hat{F} sampled after S is fixed).

If you had another way of calling Alg 3 in mind, you should completely rewrite the presentation of Alg 3 and its description. For instance, if you want to call Alg-3 k times but want to perform its first line only once, you need to separate that as a preprocessing step in another pseudo code and explicitly explain it. It sounds counterintuitive to have the same probably vector <p_1, …, p_N> and do the sampling (lines 3-6 of Alg 3) k separate times. Even if you want to do that, it takes O(N) to find this approximate function. I note that O(N) is not mentioned in the (non-preprocessing) runtime of your algorithm because you only count the number of oracle calls. Nevertheless you pay the O(N) in runtime and you should mention it.

There is a third interpretation which is calling Alg 3 only once and using the same approximate function in all k iterations of greedy. I strongly recommend correcting the presentation to reflect this ambiguity.

Related work:
The problem of submodular maximization in the case of multiple underlying submodular functions has been studied previously in the context of probabilistic submodular maximization or two stage submodular maximization.
Probabilistic Submodular Maximization in Sub-Linear Time, Stan et al. ICML 2017
Learning Sparse Combinatorial Representations via Two-stage Submodular Maximization
Eric Balkanski et al. ICML 2016.

In this setting, the submodular function we want to optimize is sampled from a distribution (e.g. each user has a submodular valuation function and a user is drawn from the distribution). Their approach is to compute a compact subset of items in the ground set and study the compression and approximation errors separately. These methods are not necessarily directly applicable to your setting but doing a more thorough compare and contrast seems necessary. In particular, their results are constant approximations strictly below the 1-1/e target approximation in your paper. I should mention that this connection is different from the relevance of your work to Mini-batch methods you reviewed in the related work section.

Typos:
1. The equation after theorem 1: \hat{F}^j_{S_j} → \hat{F}^j_{S_j}(e)
2. Last inequality in page 4: The lower bound on F(S_{k+1}) should have the extra error term, gamma epsilon’ F(S^*) as a deductive term not addition. The current form is a stronger guarantee even though we have the additive error term. I believe it should be:
F(S_{k+1}) \geq F(S^*)\beta - \gamma \epsilon’ F(S^*)
3. Page 5, Mini-batch setting paragraph: … its is … → it is

**Questions:**

Their algorithm becomes better than the naive approach if N is larger than n times k where n is the number of items in the ground set and k is the cardinality constraint.

Question to authors: Could you provide a practical example that N is substantially larger than n times k to motivate your algorithms?

Approximation of function F:
In line 6 of algorithm 2, you look for an approximate version of the function F_{S_j}. The description of how you use 3 to achieve that is not completely clear. But the likeliest interpretation that comes to mind is the following. You call algorithm 3 with function F initialized as F_{S_j} in the j’th iteration of the Greedy algorithm. This requires O(Nn) oracle calls because of line 1 of Alg 3 every time you call Alg 3. Since you need k approximate functions for the k iterations of the greedy algorithm, overall you have O(Nnk) oracle calls which makes the whole motivation point of your paper irrelevant.

The statement of Lemma 8 suggests that you had the above interpretation in mind (\hat{F} sampled after S is fixed).

If you had another way of calling Alg 3 in mind, you should completely rewrite the presentation of Alg 3 and its description. For instance, if you want to call Alg-3 k times but want to perform its first line only once, you need to separate that as a preprocessing step in another pseudo code and explicitly explain it. It sounds counterintuitive to have the same probably vector <p_1, …, p_N> and do the sampling (lines 3-6 of Alg 3) k separate times. Even if you want to do that, it takes O(N) to find this approximate function. I note that O(N) is not mentioned in the (non-preprocessing) runtime of your algorithm because you only count the number of oracle calls. Nevertheless you pay the O(N) in runtime and you should mention it.

There is a third interpretation which is calling Alg 3 only once and using the same approximate function in all k iterations of greedy. I strongly recommend correcting the presentation to reflect this ambiguity.

Question: which of the above interpretations did you have in mind?

Typos:
1. The equation after theorem 1: \hat{F}^j_{S_j} → \hat{F}^j_{S_j}(e)
2. Last inequality in page 4: The lower bound on F(S_{k+1}) should have the extra error term, gamma epsilon’ F(S^*) as a deductive term not addition. The current form is a stronger guarantee even though we have the additive error term. I believe it should be:
F(S_{k+1}) \geq F(S^*)\beta - \gamma \epsilon’ F(S^*)
3. Page 5, Mini-batch setting paragraph: … its is … → it is

---

> ### Author Response · Authors · 2023-11-12
>
> Thank you for carefully reading our paper and for the constructive feedback! Please allow us to address your main concerns here first, and implement them in a revision later.
>
> > Could you provide a practical example that N is substantially larger than n times k to motivate your algorithms?
>
> Indeed, we did not provide proper motivation for our work. Thank you for pointing this out.
>
> As noted by Rafiey and Yoshida the most natural example is welfare maximization. We provide an intuitive explanation below:
>
> Imagine you are tasked with deciding on a meal menu for a large group of N people (e.g., all students in a university, all high school students in a country). You need to choose k ingredients to use from a predetermined set (chicken, fish, beef, etc...) of size n. Every student has a specific preference, modeled as a monotone submodular function $F_i$. Our goal is to maximize the welfare of all students: $\sum_i^N F_i$. Note that in this case, N is much larger than n times k.
>
> The standard algorithm will require asking every student k questions. That is, you would start by asking all students: "Rate how much you would like to see food X on the menu" for n possible options. You would then need to wait for all of their replies, and continue with "Given that chicken is on the menu, rate how much you would like to see food X on the menu" for n-1 possible options and so on. Even if you do an online poll, this is still very time-consuming (as you will need to wait for everyone to reply in every step, repeating for k steps).
>
> Using the sparsifier approach of Rafiey and Yoshida the preprocessing step is not practical as it will require asking every student poly(N) questions. Furthermore, the dependence on B means that you might end up having to poll all of the students anyway.
>
> On the other hand, our approach only requires every student to answer how much he likes any specific food (chicken, beef, fish, etc...). Then for k steps, we poll a much smaller set of students asking them which food they would like to see added to the menu.
>
> A few more points:
> - In this example, it is also clear that the computation time (deciding on the subset of students to poll at every step) is negligible compared to the query time (asking for their opinion and waiting for them to answer).
>
> - If there is a set of constraints, and they suddenly change, we do not need to perform the preprocessing step again.
>
> - While this might seem like a toy example, welfare maximization is an extremely important and useful problem. It is easy to see that the above translates to much more important scenarios, like allocating public resources (e.g., including medical treatments in a medical insurance program, designing a curriculum of classes in a school, etc...). A more CS-oriented example might be adding features to an online service such as to maximize user utility.
>
> > Approximation of function F:
>
> Indeed, what we meant was that line 1 of Alg 3 is called only once as a preprocessing step and then Algorithm 3 is called in every iteration of Alg 2 to get an approximate version of the function $F_{S_j}$. You are correct in noting that the running time of Alg 3 is $O(N)$. So the total running time of the algorithm (not oracle calls) depends on N. It is standard in the literature to take oracle calls as the main complexity measure of the algorithm. But we completely understand your point. We will add this point to our paper and clarify the presentation.

---

> > ### Comment · Reviewer_qFjC · 2023-11-15
> > **Example provided to support the motivation of the setting**
> >
> > The example provided in the rebuttal helps a bit the case that the number of users is much larger than other input factors. In light of that I am changing my score from reject to weak reject. I still don't think this paper qualifies to be accepted in ICLR.

---

### Official Review · Reviewer_ev1Q · 2023-10-28

**Soundness:** 2 fair
**Presentation:** 2 fair
**Contribution:** 2 fair
**Rating:** 5
**Confidence:** 4

**Summary:**

This paper presents a mini-batch algorithm for maximizing a non-negative decomposable submodular function under a set of constraints. The algorithm reduces the number of oracle evaluations compared to previous methods and achieves good approximation guarantees. The paper provides a proof for the algorithm's effectiveness and compares it to other algorithms. The algorithm is shown to be effective for cardinality and p-system constraints. Overall, the paper contributes a new algorithm that efficiently optimizes submodular functions with good approximation guarantees.

**Strengths:**

The paper introduces a mini-batch algorithm that maximizes non-negative decomposable submodular functions under constraints, which reduces the number of oracle evaluations in some cases. The methodology is sound and the results are well-supported. Furthermore, the paper underscores the potential for applying the mini-batch approach to other constraints and functions beyond submodularity, indicating its broader applicability.

**Weaknesses:**

1.	The paper posits that the time complexity of the naïve greedy algorithm is dependent on N, which leads to high time complexity when N is large. The motivation behind their algorithm is to eliminate this dependence on N. It would be beneficial for the authors to provide a substantial number of examples demonstrating scenarios in real life where N is exceptionally large. This would strengthen their motivation, especially considering that there are few published papers with a similar motivation, indicating that such a motivation has not yet been widely accepted.
2.	The results obtained by the paper under the unbounded curvature case do not seem to significantly improve upon the results of Rafiey & Yoshida (2022). The superiority or inferiority between the two depends on the values of B, n, p, and k.
3.	The paper lacks experiments to validate the performance of the proposed algorithm. It would be advantageous for the authors to include empirical evidence supporting their theoretical claims.
4.	The organization of the paper appears to be chaotic, giving an impression of an unprepared manuscript rather than a ready-to-publish paper.

**Questions:**

Please refer to weaknesses.

---

> ### Author Response · Authors · 2023-11-12
>
> Thank you for taking the time to read our paper! Please allow us to address your main concerns here first, and implement them in a revision later.
>
> 1) Please refer to our reply to Reviewer qFjC.
>
> 2) Our main contribution is removing the dependence on B. In doing so we improve the worst case query complexity from exponential to polynomial in n. Indeed when k or p are very large and B is constant our results do not improve over Rafiey & Yoshida. However, it is often the case that $k,p \ll n$ (e.g, for submodular maximum matching on dense graphs $n = |V|^2, p=2, k=|V|$), and it is not clear why B should be small in general.
>
> 3) We believed that a theoretical result presenting an exponential improvement in query complexity was sufficient. Clearly, we were mistaken. May we ask the reviewer what would be a good experimental setup for our case? Specifically, which dataset would you suggest, and how would you compare the query complexity between a mini-batch algorithm and a sparsifier?
>
> 4) We would be more than happy to improve the readability of our paper! May we ask for concrete issues that we can improve upon?

---

> > ### Comment · Reviewer_ev1Q · 2023-11-16
> >
> > I would suggest that the authors provide a more detailed discussion on the contributions of this paper to the machine learning community, including some specific application examples and experimental results. This is particularly important considering that the paper addresses non-traditional submodular maximization problems, which may not be well-known in the machine learning community. Additionally, I would recommend that the authors dedicate a section specifically to formally defining the problem studied in the paper and introducing other relevant symbols.
> >
> > These are just some friendly suggestions from my personal perspective.

---

### Official Review · Reviewer_dUP7 · 2023-10-31

**Soundness:** 3 good
**Presentation:** 3 good
**Contribution:** 2 fair
**Rating:** 3
**Confidence:** 4

**Summary:**

The paper studies the problem of maximizing a decomposable monotone submodular function $F$ over a ground set of size $n$. By decomposable it means that $F=\sum_{i=1}^N f_i$, where each $f_i$ is monotone submodular and can be accessed via a value oracle. The paper presents fast algorithms for the problem under the cardinality constraint and the $p$-system constraint. More specifically, for the cardinality constraint, a naïve greedy algorithm needs $O(Nnk)$ queries, where $k$ denotes the constraint parameter. In contrast, the paper presents an algorithm using $O(Nn+k^3n^2/\epsilon^2)$ queries up to a logarithmic term. Furthermore, when the function has a bounded curvature, the algorithm uses $O(Nn+kn^2/\epsilon^2)$ queries up to a logarithmic term. This improves over two previous results. For the $p$-system constraint, a similar improvement can be obtained.

**Strengths:**

1.	The basic idea for solving the problem is to construct a sparsifier of $F$ which approximates $F$ at all subsets. The paper observes that one can construct a sparsifier which only approximates the marginal values of each single element during the execution of the greedy algorithm. The observation is clever and general. I believe it can be applied for more general functions and various constraints.
2.	The algorithm and analysis are presented in a transparent and readable way.

**Weaknesses:**

1.	The motivation of the paper is problematic. Since the problem itself is a traditional combinatorial optimization problem, it is strange to consider the preprocessing part. In my understanding, the preprocessing part is meaningful if we consider one decomposable submodular function under a large number of different constraints, since the preprocessing part is not related to the constraint. But I am not sure about the motivation of this kind of setting. Besides, even in the case where the preprocessing part is necessary, the proposed algorithm is superior to the naïve $O(Nnk)$ algorithm only when $N$ is large. However, since the Nn term looks inevitable, I suspect that for large $N$, the proposed algorithm is still too slow in practice. Unluckily, the authors do not explain more about this. Specifically, the authors do not suggest any applications where $N$ is in fact large and the paper lacks experiments to demonstrate the advantage of their algorithm for solving such instances.
2.	The paper has few technical contributions. It builds on a simple observation. Although this observation is clever, it is more like a trick and does not need many insights into the problem.
3. The proof of Theorem 3 is problematic. Note that the sparsifier is determined before the greedy algorithm is executed. So it’s unknown which $S_j$’s the greedy algorithm will reach. Consequently, to ensure that the incremental value of $S_j$ can be estimated accurately, one needs to combine the union bound with Lemma 8 to show the estimate is accurate for all subsets of size at most $k$. This means that $\alpha$ should be $\Theta(\epsilon^2 k \log n)$ and now the number of oracle evaluations is $O(k^2n^2 \log n)$. So, under the cardinality constraint, the proposed algorithm has no advantage over Kudla & Zivny (2023), which further reduces the contribution of the paper.
4.	The paper is presented in a TCS rather than an ML style, which makes it less attractive to AI researchers. I suggest the authors to rewrite the Introduction and explain why the paper’s results are important to the AI community.

**Questions:**

1.	The sentence “The expected number of oracle evaluations of our algorithm only depends on the size of the ground set” in the Abstract is misleading. I understand the authors regard the Nn term as the preprocessing time. However, it will not disappear in practice. And it’s unfair to compare with previous algorithms while omitting this term.
2.	Rafiey & Yoshida (2022) is published in AAAI22, please update the reference.
3.	I suggest the authors to moving the Related Work to the beginning of the Introduction, and explain more about the reason why the paper’s results are important to the AI community.
4.	Page 2, in the second to last paragraph, I can not figure out why the sentence “While the mini-batch approach results…” can imply the sentence “This means that we need to reestablish…”.

---

> ### Author Response · Authors · 2023-11-12
>
> Thank you for the valuable feedback! Please allow us to address your main concerns here first, and implement them in a revision later.
>
> 1) Please refer to our reply to Reviewer qFjC for motivation regarding $N \gg n,k$. We are not sure we understand your concern regarding the preprocessing time. Our motivation is exactly the same as Rafiey & Yoshida. Specifically, assume you would like solve the problem over many different constraints, you would only need to perform the preprocessing step once. This is why we follow the approach of Rafiey & Yoshida and account for it separately. Please let us know if we misunderstood what you meant here.
>
> 2) Indeed our solution is technically very simple, and our main contribution is a different perspective overlooked by previous papers. Nevertheless, this provides an exponential improvement in the worst-case query complexity which we believe to be a significant result. The significance of our results is of course subjective, but wouldn't you agree that if the result is significant then simpler is better (e.g., the 2 page mini-batch k-means paper by Sculley)?
>
> 3) We are not sure we understand your concerns here. At every iteration of the greedy algorithm we sample a "fresh" sparsifier (i.e., mini-batch) after $S_j$ is fixed before we sample the mini-batch for each iteration and therefore we only need to union bound over all possible extensions of $S_j$ by one element. As Reviewer qFjC pointed out, it might be the case that our algorithm description is not sufficiently clear, and we will improve this in the revision.
>
> 4) Indeed, we are theoreticians :) As submodular maximization is well studied in the ML community, do you mean we should elaborate more on the importance of decomposable submodular functions? If so, please see our reply to Reviewer qFjC. If not, could you please provide more details on changes you would like to see made?
>
> >The sentence “The expected number of oracle evaluations of our algorithm only depends on the size of the ground set” in the Abstract is misleading. I understand the authors regard the Nn term as the preprocessing time. However, it will not disappear in practice. And it’s unfair to compare with previous algorithms while omitting this term.
>
> We understand the concern, please see our response to question (1). Anyway, we will change the wording in the abstract.
>
> >Page 2, in the second to last paragraph, I can not figure out why the sentence “While the mini-batch approach results…” can imply the sentence “This means that we need to reestablish…”.
>
> What we meant to say here is that although our approach has better query complexity computing a sparsifier has the benefit that you do not need to reprove the approximation  guarantees of known results.

---

> > ### Comment · Reviewer_dUP7 · 2023-11-20
> >
> > After reading the example in the  reply to Reviewer qFjC, I think the motivation to consider the case N>>n is interesting and meaningful. I still suggest the author to thoroughly discuss the motivation of the whole setting in the paper, not only the importance of decomposable submodular functions, but also the motivation of the case N>>n and the case where the preprocessing only runs once but the optimization runs several times.

---

### Official Review · Reviewer_QSfG · 2023-10-31

**Soundness:** 3 good
**Presentation:** 2 fair
**Contribution:** 3 good
**Rating:** 3
**Confidence:** 3

**Summary:**

In this paper, the authors present a new algorithm for maximizing monotone decomposable submodular functions subject to cardinality and p-system constraints.

**Strengths:**

1. The improvement in preprocessing time and the main algorithm for unbounded curvature is beneficial when $N \gg n$.
2. Removing the dependence on $B$ is also beneficial in the bounded curvature setting.
3. Although "mini-batching" ideas have been used in this context before, the way they are used in this work is novel.

**Weaknesses:**

1. The paper is poorly written and difficult to understand in some places. Algorithm 2 is not an algorithm, line 4 does not make sense, and line 7 at least needs a pointer.
2. The paper does not provide sufficient justification for the importance of the problem or their algorithms improvements.
3. The paper does not compare the results in an experimental setting, which, combined with the above issue, raises significant concerns about the applicability of the authors' algorithms.
4. The authors claim that "The expected number of oracle evaluations of our algorithm only depends on the size of the ground set," which is clearly impossible. Their preprocessing still has linear dependency on N. I believe this type of overreaching claims is very confusing and can give the false impression of the quality of the work.

**Questions:**

Please provide responses for the above concerns.

---

> ### Author Response · Authors · 2023-11-12
>
> Thank you for your comments.
>
> 1) Please see our response to Reviewer qFjC.
> 2) Please see our response to Reviewer qFjC.
> 3) Please see our response to Reviewer ev1Q.
> 4) Please see our response to Reviewer dUP7.

---

> > ### Comment · Reviewer_QSfG · 2023-11-15
> > **Response to Authors**
> >
> > After reading the authors response and other reviewers comments I think:
> >
> > 1- The presentation of the paper should be significantly improved.
> > 2- The importance of the problem and its applications is not clear. Authors explanations are not convincing.
> > 3- The contribution of the work is below the expectations for this conference.
> >
> > I updated my score accordingly.